

# Comparing the ice nucleation properties of the kaolin minerals kaolinite and halloysite

Kristian Klumpp, Claudia Marcolli, Ana Alonso-Hellweg, Thomas Peter

[1] Institute for Atmospheric and Climate Sciences, ETH Zurich, Zurich, 8092, Switzerland

*Correspondence to*: Kristian Klumpp (kristian.klumpp@env.ethz.ch) and Claudia Marcolli (Claudia.marcolli@env.ethz.ch)

**Abstract.** Heterogeneous ice nucleation on dust particles in the atmosphere is a key mechanism for ice formation in clouds. However, the conditions of a particle surface for efficient ice nucleation are poorly understood. In this study we present results of immersion freezing experiments using differential scanning calorimetry on emulsified mineral dust suspensions, involving the two chemically identical, but morphologically different kaolin minerals kaolinite and halloysite. Kaolinite occurs in a platy

morphology, while halloysites form predominantly tubular structures. We investigated six different halloysite and two different kaolinite samples. Our results show that, on average, the halloysite samples exhibit a higher ice nucleation (IN) activity, than the kaolinite samples, but also a higher diversity in terms of freezing onset temperatures and heterogeneously frozen fraction. Repeating the freezing experiments after shortly milling the samples led to a decrease in freezing onset temperatures and in the heterogeneously frozen fraction of the halloysite samples, bringing their IN activity closer to that of the kaolinites. To

interpret these findings, the freezing experiments were complemented by dynamic vapour sorption (DVS) measurements, pore ice melting experiments with slurries, and transmission electron microscopy (TEM) before and after milling. These measurements demonstrate the destruction of tubes by milling and provide evidence for the influence of the tubular structure of the halloysites on their IN activity. We identify the OH–Al–O–Si–OH functionalized edges as the most likely site for ice nucleation, as the high geometric diversity of the edges best accounts for the high diversity in IN activity of halloysites. We

hypothesize that the stacking of layers and the number of stacks in halloysite tubes and kaolinite platelets affect the freezing temperature, with thicker stacks having the potential to freeze water at higher temperatures. The notion that the edges constitute the IN-active part of kaolin minerals is further supported by comparing kaolin minerals with montmorillonites and feldspars, all of which exhibit enhanced IN activity in the presence of ammonia and ammonium-containing solutions. As OH–Al–O–Si–OH functionalized edge surfaces are the only surface type kaolin particles have in common with montmorillonites and

feldspars, the common feature of IN activity enhancement in ammoniated solutions can only be explained by ice nucleation occurring at the edges of kaolin minerals.

## 1. Introduction

The amount of ice in a cloud has a large effect on various cloud properties, including cloud lifetime, precipitation formation, radiative properties and chemical processes (Lohmann, 2006; IPCC, 2013; Field and Heymsfield, 2015; Mülmenstädt et al.,

2015). There are two basic types of ice nucleation (IN), namely homogeneous and heterogeneous. Homogeneous nucleation





occurs in supercooled cloud droplets at temperatures between 238 K and 235 K, depending on droplet size (Sassen and Dodd, 1987; Heymsfield and Sabin, 1989). Solution droplets can cool to even lower temperatures due to the freezing point depression (Koop et al., 2000). In heterogeneous ice nucleation so-called ice nucleating particles (INPs) are involved, which lower the energy barrier that must be overcome in order to form an ice embryo, increasing the freezing temperatures to values between

the homogeneous freezing temperature and the melting point. Depending on the considered interplay between the INP and the cloud droplet, different freezing modes are distinguished. Immersion freezing describes the freezing process caused by an INP fully immersed in a cloud droplet. Condensation freezing is assumed to take place when ice nucleation coincides with cloud droplet activation (Vali et al., 2015; Kanji et al., 2017). Contact freezing occurs when the INP initiates freezing by penetrating the surface of a droplet (Durant and Shaw, 2005; Shaw et al., 2005; Nagare et al., 2016). Deposition nucleation describes ice

formation directly from water vapor to a solid ice crystal without involving the liquid phase. Deposition nucleation occurring in the absence of liquid water is currently questioned by Marcolli (2014), who suggested condensation of water vapor in pores of INPs with subsequent freezing (pore condensation and freezing; PCF) as the relevant nucleation path for ice nucleation below water saturation.

Many different particle types are considered relevant INPs, including a wide chemical diversity. Mineral dust, soot, biological

material (bacteria, fungal spores, pollen, diatoms) and organic compounds are frequently mentioned (Hoose und Möhler, 2012; Kanji et al., 2017). There is increasing evidence that ice nucleation does not occur on the entire surface of INPs but only at preferred sites called nucleation sites, while the rest of the surface remains inactive (Vali, 2008; 2014; Vali et al., 2015). Evidence for such sites stems from the observation of INPs during repeated freeze-thaw cycles. In such refreeze experiments nucleation shows non-stochastic behaviour with sudden jumps in freezing temperature from one cycle to the next (Wright and

Petters, 2013; Kaufmann et al., 2017), a behaviour that would not be expected if the whole surface area of the particle were involved. Moreover, when quartz and feldspar surfaces were subjected to repeated freeze-thaw cycles in immersion freezing experiments, only a very limited number of sites were observed to induce ice nucleation (Holden et al. 2021). Similarly, when aqueous solutions of the protein apoferritin underwent repeated freeze-thaw cycles in a multiwell tray, the same wells tended to freeze among the first during all cycles (Cascajo-Castresana et al., 2020).

The properties that distinguish nucleation sites from the rest of the surface are still only poorly understood. While there is compelling evidence that pores are required for ice nucleation at relative humidity (RH) below water saturation (Christenson, 2013; Marcolli, 2014; 2020; Campbell et al., 2017; Campbell and Christenson, 2018; David et al., 2019; 2020; Marcolli et al., 2021), the requirements for sites that induce ice nucleation in immersion mode are less well understood. Classical nucleation theory (CNT) can be used to infer the critical size required for a site to host an ice embryo. By applying CNT parameterizations

to diverse types of INPs, Kaufmann et al. (2017) derived site areas of 10–50 nm$^2$ assuming flat nucleation sites. Apart from size, also chemical and topographical requirements are considered relevant for IN activity. Crystal match is often discussed as a main prerequisite enabling a surface to accommodate ice. For example, the high IN activities of silver iodide and of 2D crystals formed by long-chain alcohols have been associated with close lattice match to ice (Popovitz-Biro et al., 1994;



Majewski et al., 1995; Cantrell and Robinson, 2006; Zobrist et al., 2007; Knopf and Forrester, 2011; Marcolli et al., 2016).
Yet, molecular simulations of ice nucleation on silver iodide surfaces showed that chemical surface properties are even more relevant than lattice match, since only the faces exposing silver ions were found capable to template ice (Zielke et al., 2015). Moreover, surfaces without lattice match to ice and even macromolecules proved good ice nucleators, shedding doubt on the relevance of crystallographic properties for ice nucleation (DeMott, 1990; Diehl et al., 1998; DeMott et al., 1999; Knopf et al., 2010; Murray et al., 2010; Wang et al. 2012; Wilson et al., 2012). Rather, the ability of the surface to form hydrogen bonds
with water molecules and topographical properties have gained increasing attention as relevant features of nucleation sites. In this regard, Pedevilla et al. (2017) showed in a molecular dynamics study, that it is not the arrangement of hydroxyl groups in a particular pattern, but their surface density and the strength of substrate-water interaction that are relevant descriptors of IN activity. Nevertheless, even for the highly hydroxylated surface of quartz, IN activity is restricted to scarce nucleation sites that increase in abundance when defects are introduced through milling (Zolles et al., 2015; Kumar et al., 2019a). The relevance
of surface geometry has been pointed out in a molecular dynamics study by Bi et al. (2017), who found that the IN activity of a wedge depends on its opening angle.

Numerous laboratory and modelling studies have attempted to elucidate the origin of IN activity of mineral dusts. The IN activity of feldspars has been related to the preferential exposure of specific crystal faces (Kiselev et al., 2017; 2021; Pach and Verdaguer, 2019), to microtexture (Whale et al. 2017), and to the specific surface cations (Zolles et al., 2015; Yun et al., 2021).
Furthermore Kumar et al. (2022) showed the relevance of stacking and thickness of smectite tactoids for IN activity and freezing temperatures, while impurities and exchangeable cations were only relevant through their influence on stacking and delamination. Another mineral whose ability to nucleate ice has been investigated in many experimental and modelling studies is kaolinite. Kaolinite is a common clay mineral found in airborne mineral dust close to source regions and in even higher proportions in transported dust. Because clay minerals are present in the fine particle fraction, they can be transported over
longer distances compared to minerals that are associated with the coarse fraction, such as quartz (Gomes et al., 1990; Reid et al., 2003; Vlasenko et al., 2005). Kaolinite forms platelets exposing on one side a hydroxylated Al surface with Al atoms arranged in a hexagonal pattern and on the other side a siloxane surface with Si–O–Si bridges forming hexagonal rings, which do not swell upon wetting (Bear, 1965). Several modelling studies have shown that the hydroxylated Al surface of kaolinite is capable of growing the primary prism face of ice on top of it (Zielke et al., 2016; Sosso et al., 2016; Glatz and Sarupria, 2018;
Soni and Patey, 2021). They ascribed the IN activity of this surface to its lattice match to ice together with favourable hydrogen bonding with water molecules.

As a common mineral found in airborne mineral dusts (Murray et al., 2012), the IN activity of kaolinite has been investigated in numerous experimental studies. A study conducted by Hoffer (1961) yielded median freezing temperatures of 240 K (ranging from 238 to 254 K) in water droplets (100–170 µm diameter) containing kaolinite particles. Experiments from Pitter
and Pruppacher (1973) conducted in a wind tunnel setup with levitated kaolinite suspension droplets (650 µm diameter) yielded freezing temperatures between 243 and 259 K. Both studies do not state the concentration or mass of mineral dust that was





used in these experiments. This wide freezing range is strongly narrowed when the IN activity of individual particles rather than of large particle ensembles is investigated. Studies on single, size-selected kaolinite particles performed in immersion mode yielded freezing onsets in the range of 240–244 K depending on particle size (Lüönd et al., 2010; Wex et al., 2014; Nagare et al., 2016). Emulsion freezing experiments performed with differential scanning calorimetry (DSC) yielded freezing onsets around 240 K, while bulk samples measured with the same instrument exhibited freezing temperatures up to 260 K (Pinti et al., 2012; Kaufmann et al., 2016). Not all kaolinite particles proved to be IN active in these experiments. Particles with sizes below 800 nm did not reach frozen fractions of unity before homogeneous ice nucleation set in (Lüönd et al., 2010; Wex et al., 2014; Nagare et al., 2016), which is in accordance with parameterizations of active site densities as a function of kaolinite surface area (Wex et al., 2014; Hartmann et al., 2016). Moreover, time-dependent immersion freezing experiments on size-selected particles were in accordance with the assumption that ice nucleation occurred on nucleation sites, which promote ice nucleation in a characteristic temperature range with nucleation rates that increase steeply as temperature decreases (Welti et al., 2012; Marcolli et al., 2007). However, the presence of nucleation sites is not consistent with molecular dynamics studies, which suggest that ice nucleation can occur anywhere on the hydroxylated alumina surface of kaolinite (Zielke et al., 2016; Sosso et al., 2016; Glatz and Sarupria, 2018; Soni and Patey, 2021). Therefore, the question is whether this surface is capable of promoting ice nucleation sufficiently to compete with homogeneous nucleation under typical environmental or experimental conditions, when small ice-nucleating surfaces face large volumes of water. To become an efficient nucleation site, the presence of additional topographic or structural elements might be required to augment the IN activity derived for the regular kaolinite surface. In this study, we explore this hypothesis by comparing the IN activity of kaolinite with that of halloysite. Like kaolinite, halloysite belongs to the kaolin mineral group and has the same chemical composition and crystal structure as kaolinite. Yet, instead of forming flat platelets, it appears in a variety of different morphological structures. Spherical, platy particle shapes, as well as short tubular and elongated tubular morphologies have been documented (de Souza Santos et al., 1964; Dixon and McKee, 1974; Noro, 1986; Singer et al., 2004; Joussein et al., 2005), although halloysite occurs mainly in an elongated tubular morphology commonly referred to as halloysite nanotubes (HNTs).

Even though halloysite has been found in rainwater and snow samples (Ishizaka, 1972, 1973; Kumai, 1976), it has been much less studied than kaolinite with respect to ice nucleation. Hoffer (1961) conducted freezing experiments with halloysite suspensions but found no difference in the IN properties compared to the also measured kaolinite samples.

This study presents the results of immersion freezing experiments with emulsified halloysite and kaolinite suspensions in a differential scanning calorimeter (DSC). We examined six different halloysite samples, two different kaolinite samples, and additionally, samples of both minerals after they have been milled. Furthermore, we supplement the DSC measurements with dynamic vapour sorption (DVS) analysis and transmission electron microscopy (TEM) to further characterize the differences between the samples.



## 2. Mineralogy

### 2.1 Kaolinite

Kaolinite is a non-swelling 1:1 clay mineral with the chemical composition of $Al_4Si_4(OH)_8O_{10}$. It forms T–O layers consisting of alternating silicon oxide tetrahedral (T) and aluminum oxide octahedral (O) sheets, which are connected by shared O atoms. In the regular crystal structure, there are no exchangeable ions, leaving only defects and edges susceptible towards ion exchange. Hydrogen bonds between Al–OH and Si–O–Si provide attraction between the sheets and prevent small molecules and ions from entering the interlayer region (Deer et al., 1992). Measurements conducted on the two standard kaolinites KGa-

1b and KGa-2 gave layer thicknesses of 7.169 Å and 7.184 Å, respectively (Hillier et al., 2016). Kaolinite particles consist of multiple layers stacked on top of each other. This leads to the formation of platy particles, ranging typically from 0.3 to 4 µm in lateral dimension and from 0.05 to 2 µm in thickness (Schwertmann, 1966). Cleavage of kaolinite occurs along the basal planes, resulting in the formation of an Al–OH surface on one side and a siloxane (Si–O–Si) surface on the other side. The remaining surface consists of the alternating sheet boundaries, which are orthogonal to the basal planes, usually referred to as

the "particle edges".

In this study, we investigate the two above mentioned kaolinite samples KGa-1b and KGa-2 (subsequently also termed K1 and K2, respectively), which are provided by the Clay Minerals Society (see https://www.clays.org/). KGa-1b as well as KGa-2 are reported to have a very similar composition of 96 % kaolinite, 3 % anatase and 1 % candrallite with impurities of dickite and quartz in KGa-1b and impurities of mica and/or illite in KGa-2 (Chipera and Bish, 2001). KGa-1b has higher crystallinity

than KGa-2 (Zhang et al., 2003; Soro et al., 2003) and is therefore labelled as "low defect kaolinite" by the Clay Minerals Society (i.e., "well-ordered" kaolinite KGa-1b and "poorly-ordered" kaolinite KGa-2).

### 2.2 Halloysite

As kaolinite, halloysite is a 1:1 clay mineral composed of layers consisting of a tetrahedral silica and an octahedral alumina sheet. However, instead of flat platelets, halloysite typically occurs in the form of cylindrical tubes, but can also form

spheroidal, platy or tabular particles (Churchman et al., 1995; Joussein et al., 2005). One reason these morphologies differ from those of kaolinite is that when halloysite formed, monolayers of water molecules were interlaced between its unit layers (Joussein et al., 2005; Yuan et al., 2015). Yet, the interlayer water is not strongly bound, and halloysite is therefore readily dehydrated (Joussein et al., 2005). Indeed, mild heating, low pressures or low RH are sufficient to lead to irreversible dehydration of the mineral (Schwertmann, 1966; Kohyama et al., 1978a; Joussein et al., 2005), making hydrated halloysite

being rare and difficult to handle. Accounting for the interlaced water layer, the chemical formula is $Al_4Si_4(OH)_4O_{10} \cdot nH_2O$ with $n$ varying between zero (dehydrated) and two (fully hydrated). Fully hydrated halloysite has an interlayer spacing of 10 Å and is therefore often referred to as 10 Å-halloysite. As the basal spacing in dehydrated halloysite collapses to about 7 Å, it is often referred to as 7 Å-halloysite. More exactly, upon dehydration the basal spacing of halloysite does not altogether reach values of kaolinite but remains typically above 7.2 Å, a difference that is often used in XRD analysis to discriminate between





halloysites and kaolinites (Joussein et al., 2005). Halloysite tubes consist of several rolled unit layers with the siloxane surface
pointing outwards and the aluminol surface pointing inwards (Guimarães et al., 2010; Yah et al., 2012; Zhang et al., 2012).
The exact dimensions such as length, inner and outer diameter, and wall-thickness of the tubes vary depending on the source
of the sample, but also between tubes from the same source. Lengths range from 50–5000 nm; outer diameters vary between
20 and 200 nm; and inner diameters (i.e. pore widths) range from 5 to 70 nm (Pasbakhsh et al., 2013).

In this study, we focus on halloysites with tubular morphologies. We were able to procure six different halloysite samples with
this morphology. The samples Jarrahdale (subsequently termed "JA", from Western Australia), Camel Lake ("CL", South
Australia), Matauri Bay ("MB", New Zealand) and Dragonite ("DG", from the Dragon Mine in Utah, USA) were provided by
Pooria Pasbakhsh and have been characterized in Pasbakhsh et al. (2013). The commercially available sample from I-Minerals
(subsequently termed "IM", see https://imineralsinc.com/our-products/halloysite) was also provided by Pooria Pasbakhsh.
Additionally, halloysite purchased from Sigma Aldrich (termed "SA") was used. This halloysite stems from the same mine as
the DG sample. Table 1 provides an overview of morphological parameters of all samples used in this study, as well as
compositional information and their origin.

### 3. Methods

### 3.1 Immersion freezing experiments of emulsified mineral dust suspensions

The general setup of the immersion freezing experiments used here has been described in Klumpp et al. (2022) and will be
briefly repeated in this section. Immersion freezing experiments were conducted with a Differential Scanning Calorimeter
(DSC Q10 from TA instruments). All experiments were performed with emulsions freshly prepared before each DSC
measurement. Suspensions of kaolinite and halloysite with 0.2 and 1 wt % in pure water (molecular bioreagent water, Sigma
Aldrich) were prepared and sonicated for 5–10 minutes. After sonication, the suspensions were combined with a mixture of
mineral oil (93 %) and the surfactant lanolin (7 %) (both Sigma Aldrich) at a ratio of 1:4 and emulsified with a rotor stator
homogenizer (Polytron PT 1300D with a PT-DA 1307/2EC dispersing aggregate) for 40 s at 10000 rpm (for more details see
Marcolli et al., 2007; Pinti et al., 2012; and Kaufmann et. al., 2016). DSC experiments were performed with 5–10 mg of the
resulting emulsion, which were placed in an aluminum pan, which was subsequently hermetically sealed. DSC experiments
were performed at cooling and heating rates of 1 K/min. Some experiments were run with a first and third cycle performed
with a cooling rate of 10 K/min as control cycles, to test the stability of the emulsions (Marcolli et al., 2007). Experiments
were run at least twice, each time with a freshly prepared suspension. Our key parameters to analyse the DSC thermograms
are the homogeneous ($T_{\mathrm{hom}}$) and the heterogeneous ($T_{\mathrm{het}}$) freezing onset temperatures, and the heterogeneously frozen fraction
($F_{\mathrm{het}}$), which was evaluated as the integral of the heat release due to heterogeneous freezing divided by the total heat release
due to homogeneous and heterogeneous freezing as described in Klumpp et al. (2022). Occasionally occurring spikes in the
thermograms stemming from larger droplets in the droplet size distribution are excluded from the analysis.





**Table 1:** Overview of sample parameters: sample origin, specific surface area and specific pore volume (determined via $N_2$-BET and $H_2O$-DVS) from this study and from referenced literature, together with content of main mineral and main impurity.

| Sample | Sample origin | BET-surface ($H_2O$) [$m^2\,g^{-1}$] | BET-surface ($N_2$) [$m^2\,g^{-1}$] | Pore volume ($H_2O$) [$cm^3\,g^{-1}$] | Pore volume ($N_2$) [$cm^3\,g^{-1}$] | Main mineral content [%] | Main impurity |
|---|---|---|---|---|---|---|---|
| IM | I-Minerals Halloysite, Idaho (USA) | 33.4 | 28.0 [b] | 0.11 | 0.10 [b] | 93 [b] | kaolinite [b] |
| SA | Sigma Aldrich Halloysite, Utah (USA) | 59.9 | 64.4 | 0.21 | 0.13 [b] | 90 [b] | kaolinite [b] |
| DG | Dragonite, Dragon Mine, Utah (USA) | 52.6 | 57.3 [a] | 0.20 | 0.12 [a] | 84 [a] | kaolinite [a] |
| MB | Matauri Bay (NZ) | 32.6 | 22.1 [a] | 0.17 | 0.06 [a] | 87 [a] | quartz/ cristobalite [a] |
| JA | Jarrahdale (West AUS) | 42.3 | 44.6 [a] | 0.23 | 0.12 [a] | 81 [a] | quartz [a] |
| CL | Camel Lake (South AUS) | 86.9 | 74.66 [a] | 0.35 | 0.17 [a] | 95 [a] | alunite [a] |
| MB (milled) | | 77.3 | | 0.10 | | - | - |
| JA (milled) | | 75.7 | | 0.20 | | - | - |
| CL (milled) | | 82.1 | | 0.20 | | - | - |
| KGa-1b | Georgia (USA) | 10.4 | | 0.04 | | 96 [c] | anatase [c] |
| KGa-2 | Georgia (USA) | 15.7 | | 0.08 | | 96 [c] | anatase [c] |
| KGa-1b (milled) | | 15.6 | | 0.05 | | - | - |
| KGa-2 (milled) | | 27.1 | | 0.10 | | - | - |

[a] Pasbakhsh et al. (2013), [b] Hillier et al. (2016), [c] Chipera and Bish (2001)

## 3.2 Milling

Milling of clay minerals is applied to obtain materials of homogeneous fine particles. Mechanical and chemical changes during grinding of kaolinites are variable when grinding is accompanied with friction forces, such as in vibratory, oscillating, and planetary ball mills or disc mills. Generally, it is supposed that short grinding times result in reduction of kaolinite particle size and an increase in surface area, whereas prolonged grinding causes the particles to stick one to another and decreases the surface area or may result in the amorphization of crystalline structure (Valášková et al., 2011).

In the present work, we investigated the effect of short milling on heterogeneous ice nucleation. We milled the three halloysite samples CL, JA, MB and the two kaolinite samples with a tungsten carbide disk mill (Retsch RS 1, 1400 rpm) for 10 s (except for the MB sample, which was milled for 20 s). The effect of milling was investigated by means of laser diffraction analysis (particle size distribution), TEM (qualitative assessment of the structural integrity of the particles), DVS (specific surface area, pore volume), and DSC melting experiments of slurries (pore volume).



### 3.3 Dynamic vapor sorption (DVS)

DVS measurements were conducted to obtain the specific surface area (via BET analysis) and to determine the pore size of the halloysite samples. With the assumption of ideal cylindrical pores, the Cohan-Kelvin equation from Kocherbitov and Alfredsson (2007) was used to determine the pore radius $r$ to convert the adsorption isotherms to pore volume distributions:


$$r - t = -\frac{2\gamma(T)\cos\theta V_m(T)}{RT\ln(RH)} \qquad (1)$$

The thickness of the pre-adsorbed water layer in the pores $t$ can be neglected if the expected pore diameter is much larger than the width of 1–2 monolayers of water molecules (0.38 nm in Marcolli, 2014). For the surface tension $\gamma(T)$, 72 mJ m$^{-2}$ (298 K)

was used and for the molar volume of liquid water $V_m$, 18.07 ml mol$^{-1}$ (Kocherbitov and Alfredsson, 2007). The temperature of the DVS measurement was $T = 298$ K. Since the inner surface of the HNTs consists of the hydrophilic aluminol layer the contact angle $\theta$ between water and the mineral surface was approximated as $\theta = 0°$. The mass gain as a function of RH during the DVS measurement was converted to size-resolved pore volume by attributing all mass gain above 35 % RH to pore water and taking the mass increase between 35 and 98 % RH as 100 %.

### 3.4 Melting peak depression in mineral dust slurries

Slurries were prepared using the milled halloysite samples and the corresponding untreated samples. The samples CL and MB were prepared with a 4:3 ratio of dust to water (Merck; Molecular bioreagent water) and the JA sample was prepared with a 10:9 ratio. After preparation, the slurries were transferred to the DSC pans and hermetically sealed. The samples were cooled in the DSC at a rate of 10 K min$^{-1}$ until freezing occurred. The subsequent warming cycle was performed at 0.5 K min$^{-1}$ up to

275 K. Due to the melting point depression in confinement, pore water can be distinguished from bulk water depending on the pore diameter (assuming cylindrical pores). Using the following expression, we can convert the measured melting point depression ($\Delta T$) into pore radii $r$ (Faivre et al. 1999; Schreiber et al. 2001; Christenson, 2001; Jähnert et al., 2008; Kittaka et al., 2011):

$$r = \frac{2 V_m \gamma_{sl} T_{bulk}}{\Delta H_f \Delta T} \qquad (2)$$

with the interfacial tension between the solid and liquid phase $\gamma_{sl} = 31.7$ mJ m$^{-2}$ (at 273.15 K) taken from Hillig (1998), the molar melting enthalpy of water $\Delta H_f = 6.01$ kJ mol$^{-1}$ taken from Jähnert et al. (2008) and molar volume of water as described in the section above.




### 3.5 Transmission electron microscopy (TEM) imaging

Selected samples were suspended in ethanol, sonicated for five minutes and deposited on a TEM grid (Cu 200 mesh grids, Quantifoil Micro Tools GmbH, Grosslöbichau, Germany). Imaging was conducted using a JEOL-1400+ TEM (JEOL Ltd., Tokyo, Japan) operated at 120 kV. The grids have been treated prior to the imaging with a glow discharger (K100X Glow

Discharge, EM Technologies Ltd., Ashford, Kent, UK).

## 4. Results and discussion

### 4.1 Immersion freezing experiments of emulsified mineral dust suspensions

We conducted emulsion freezing experiments for all kaolinite and halloysite samples with suspension concentrations of 1 wt %

and 0.2 wt %. These two mass concentrations were chosen because of the larger surface area per unit of mass of the halloysite compared with the kaolinite samples (see Table 1 for BET surface areas) so that an overlap in the mineral surface areas is achieved across all samples. The corresponding DSC thermograms are shown in Fig. 1, and the derived $T_{het}$ and $F_{het}$ values are compiled in Fig. 2. Each sample was measured twice including fresh suspension preparation (represented by dark and light colours in Fig. 1). The smaller differences between repetitions for the higher concentrated samples suggests that the sampling

variability decreases with increasing suspension concentration. In addition to the homogeneous freezing peak at 235.4–236.6 K, all samples feature a heterogeneous freezing signal at higher temperatures. However, variations in freezing peak intensity, onset, and curve shape illustrate the differences in IN activity between the kaolinite and the halloysite samples but also between the different halloysite samples. For the two kaolinites (KGa-1b and KGa-2), heterogeneous freezing appears as broad shoulder with no clear maximum before homogeneous freezing sets in. The measurements yield average heterogeneous freezing onset

temperatures of 243.8 ± 0.5 K (0.2 wt %) and 243.3 ± 0.5 K (1 wt %) for KGa-1b and 243.6 ± 0.2 K (0.2 wt %) and 244.0 ± 0.1 K (1 wt %) for KGa-2, respectively. For KGa-1b, freezing onset ranges for both concentrations overlap, yet, with somewhat lower values for the higher concentrated sample (see Fig. 2 for individual data points). Moreover, the freezing onset temperatures of KGa-1b and KGa-2 measured in this study are higher than the values obtained by Pinti et al. (2012) when they measured the same samples with the same method in the same concentration range (0.1 to 2 wt %) about 10 years ago (KGa-

1b: 239.6–240.4 K and KGa-2: 238.6–240.2 K). We therefore tested the role of dry storage over years for the IN activity of kaolinite by suspending the sample KGa-1b (1 wt %) for one day in pure water and found the freezing onset at 240.7 K, i.e. clearly below the onset temperature of the freshly prepared samples and close to the values from Pinti et al. (2012). This indicates that dry storage indeed influences the IN activity of kaolinite towards higher freezing temperatures, an effect that is reversible when the sample is exposed to water.

**Figure 1:** Thermograms from immersion freezing experiments of emulsified kaolinite samples (uppermost two panels) and halloysite samples (lower panels). The DSC was operated at a cooling rate of 1 K min$^{-1}$. Black and grey lines: 1.0 wt % dust suspensions. Dark and light green lines: 0.2 wt % dust suspensions. The curves are normalized with respect to their total integral. Spikes occurring before the heterogeneous freezing signal originate from single large emulsion droplets and are not representative of ice nucleation by halloysite/kaolinite. Dust sample codes for kaolinites: K1 = KGa-1b ("well-ordered" kaolinite); K2 = KGa-2 ("poorly-ordered" kaolinite). Dust sample codes for halloysites: DG = Dragonite (Dragon Mine, Utah, USA); SA = Sigma Aldrich (Utah, USA); JA = Jarrahdale (Western Australia); IM = I-Minerals (Idaho, USA); MB = Matauri Bay (New Zealand); CL = Camel Lake (South Australia). See Table 1 for morphological and compositional details.

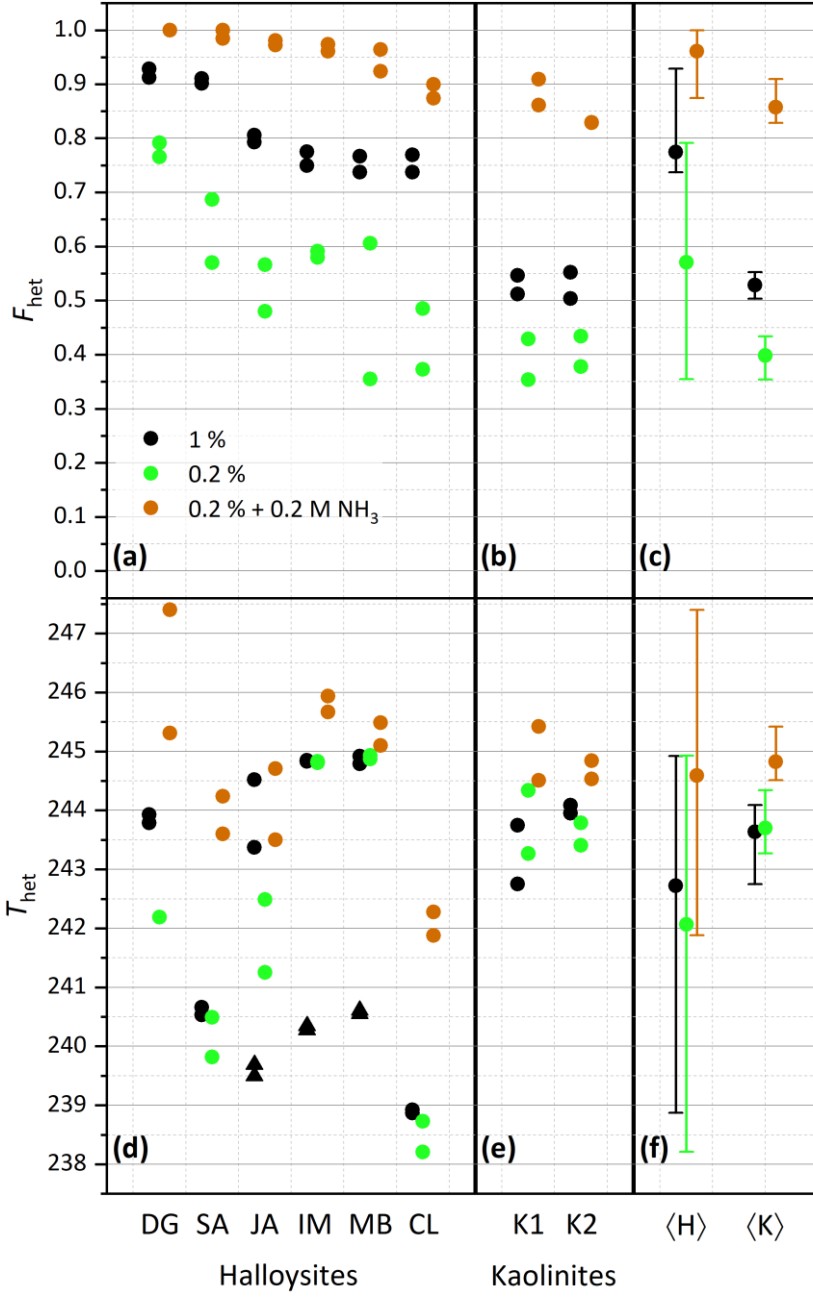

**Figure 2:** Summary of heterogeneously frozen factions $F_{het}$ (a-c) and heterogeneous freezing onset temperatures $T_{het}$ (d-f) for dust samples as obtained from the source regions without further milling. Dust sample codes as listed at the bottom of the figure are identical to those in Fig. 1. Experiments have been performed for 1 wt % (black symbols), 0.2 wt % (green) and 0.2 wt % plus 0.2 M ammonia/ammonium (brown). Triangles represent the onset of a second freezing peak when present and evaluable. Each symbol corresponds to an individual suspension preparation and DSC measurement. (a) and (d): halloysites. (b) and (e): kaolinites. (c) and (f): averages for all halloysites, ⟨H⟩, and kaolinites, ⟨K⟩, with vertical bars representing the spread (min–max) of all measurements.



After averaging over all measurements, the halloysites reveal higher $F_{het}$ than the kaolinite samples (see Fig. 2c), even if we account for the lower surface areas of the kaolinites (10–16 m$^2$/g) compared with the halloysite samples (30–90 m$^2$/g), i.e. when comparing the lower halloysite (0.2 wt %) with the higher kaolinite concentrations (1 wt %). Moreover, the variability in freezing temperatures is much larger in terms of onsets (between 238.2 K and 244.9 K) and thermogram curve shapes for

halloysites compared to the kaolinite samples. The IM and MB samples even show a double peak in the heterogeneous freezing region, which in general points towards two different types of IN active sites within a sample. Yet, both peaks lie in the temperature range observed for the other halloysite samples. Therefore, we assume halloysites to be the source of IN activity. This assumption is supported by the high mineralogical purity and the absence of IN active minerals as impurities in these samples.

Examinations of the IN activity of the halloysite samples in the presence of ammonia shed further light on the origin of the heterogeneous freezing signal. This investigation is motivated by the finding that the IN activity ($T_{het}$ and $F_{het}$) of aluminosilicates (including feldspars, kaolinite, montmorillonite and muscovite) is enhanced in the presence of ammonia or ammonium sulfate, but not the one of quartz or non-mineralogical INPs such as bacteria, fungi, and humic substances (Kumar et al. 2018; 2019a; 2019b; Worthy et al., 2021). Figure 2 indeed shows that $T_{het}$ and $F_{het}$, of 0.2 wt % suspensions are clearly

higher when prepared in 0.2 M NH$_3$ solutions compared with preparations in pure water (brown squares, corresponding thermograms can be found in the supplemental material). They are even higher than the corresponding values for the 1 wt % suspensions prepared in pure water. This indicates that the nucleation sites of the halloysites have a chemical makeup similar to those of kaolinites and aluminosilicates in general.

We investigate the role of crystal morphology for the IN activity of halloysite and kaolinite by milling the samples CL, JA,

MB, KGa-1b, and KGa-2 for a short period of time (10–20 s) and preparing 1 wt % suspensions in pure water. Figure 3 shows the measured thermograms of milled samples (blue/cyan curves) together with the original samples (black/grey curves). We observe a loss in IN activity after milling for the halloysite samples but not for KGa-1b (no significant change) and KGa-2 (increasing activity). Thus, after milling, $T_{het}$ of kaolinites and halloysites become similar, and the average $F_{het}$ of the milled halloysites falls even below the value of the kaolinites (see Fig. 4).

**4.2 Relationship between morphology and IN activity of halloysites and kaolinites**

We conducted dynamic vapor sorption (DVS) measurements to quantify BET surface areas and the pore volume of the samples before and after milling in order to elucidate the role of morphology and the morphological changes induced by milling. Furthermore, we measured the melting temperature of the pore water within the halloysite tubes in slurry experiments to constrain differences in pore volume between milled and untreated samples. Finally, we obtained TEM micrographs of the

original and milled samples to investigate the morphology before and after milling.



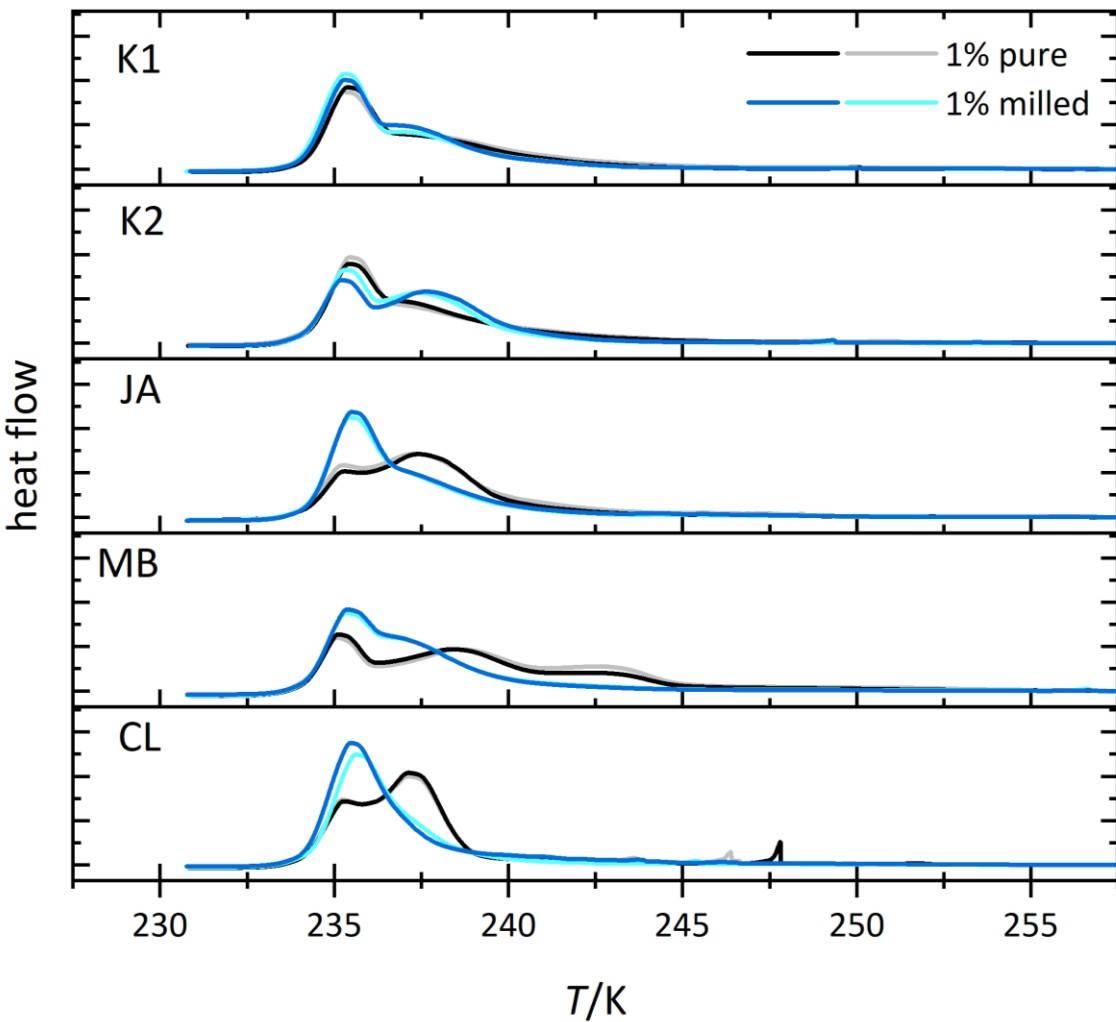

**Figure 3:** Thermograms from immersion freezing experiments of emulsified 1 wt % dust suspensions of kaolinites and selected halloysite samples (black/grey) and the corresponding measurements of the milled samples (blue/cyan). The curves are normalized with respect to their total integral. (Spikes occurring before the heterogeneous freezing signal originate from single large emulsion droplets and are not representative of ice nucleation by halloysite/kaolinite). Dust sample codes are identical to those in Fig. 1.





**Table 2:** Overview of pore parameters from this study and respective literature. Overview of pore parameters determined from DSC and DVS measurements in the present study in comparison with TEM literature data.

| Sample name | Length [nm] | Inner diameter [nm] | Outer diameter [nm] | Wall thickness [nm] |
|---|---|---|---|---|
| IM | 110–8000 840 (median)[b] | DVS: 17 | 30–410 150 (median)[b] | |
| SA | 60–2300 320 (median)[b] | DVS: 17 | 30–280 110 (median)[b] | |
| DG | 50–1500 [a] | 5–30 [a] DVS: 17 | 20–150 [a] | 5–50 [a] |
| MB | 100–3000 [a] | 15–70 [a] DVS: 20 DSC: 25–165 | 50–200 [a] | 20–100 [a] |
| JA | 50–1000 [a] | 10–30 [a] DVS: 17 DSC: 16–62 | 30–80 [a] | 10–25 [a] |
| CL | 100–1500 [a] | 10–50 [a] DVS: 10 DSC: 10–28 | 20–70 [a] | 5–30 [a] |

[a] TEM analysis by Pasbakhsh et al. (2013) (using the same samples), [b] TEM analysis by Hillier et al. (2016) (samples of same origin)

Figure 5 shows the melting thermograms of CL, JA and MB before (black/grey) and after milling (blue/cyan). While the regular melting peak at 273.15 K stems from free water of the slurry, the peak at lower temperature is due to pore water melting. This melting happens in small pores at lower temperature due to the confinement (Marcolli, 2014), and the integral of the associated melting peak is a measure for the total volume of all pores. Interestingly, for the JA and MB samples the melting peak of pore water has almost completely disappeared after milling, indicating a significant loss in pore volume. Solely in the CL sample the pore water peak is still clearly present, yet with a slight shift to higher temperatures (by 0.34 K at peak maximum).

The analysis of the DSC thermograms with respect to the onset and the maximum temperatures of the pore water melting peaks yields pore diameters of 16–62 nm for JA, 25–165 nm for MB, and 10–28 nm for CL. These ranges overlap with the inner diameter ranges derived from TEM analysis (inner diameters in Table 2; Pasbakhsh et al., 2013) yet tend to be larger than these in the case of JA and MB. The analysis of the DVS measurements with respect to relative pore volume (Fig. 6) yields maxima for pore diameters of 10–20 nm depending on the sample. These values are in the lower range of the values derived from the pore water melting peaks and TEM analysis by Pasbakhsh et al. (2013; inner diameters in Table 2).




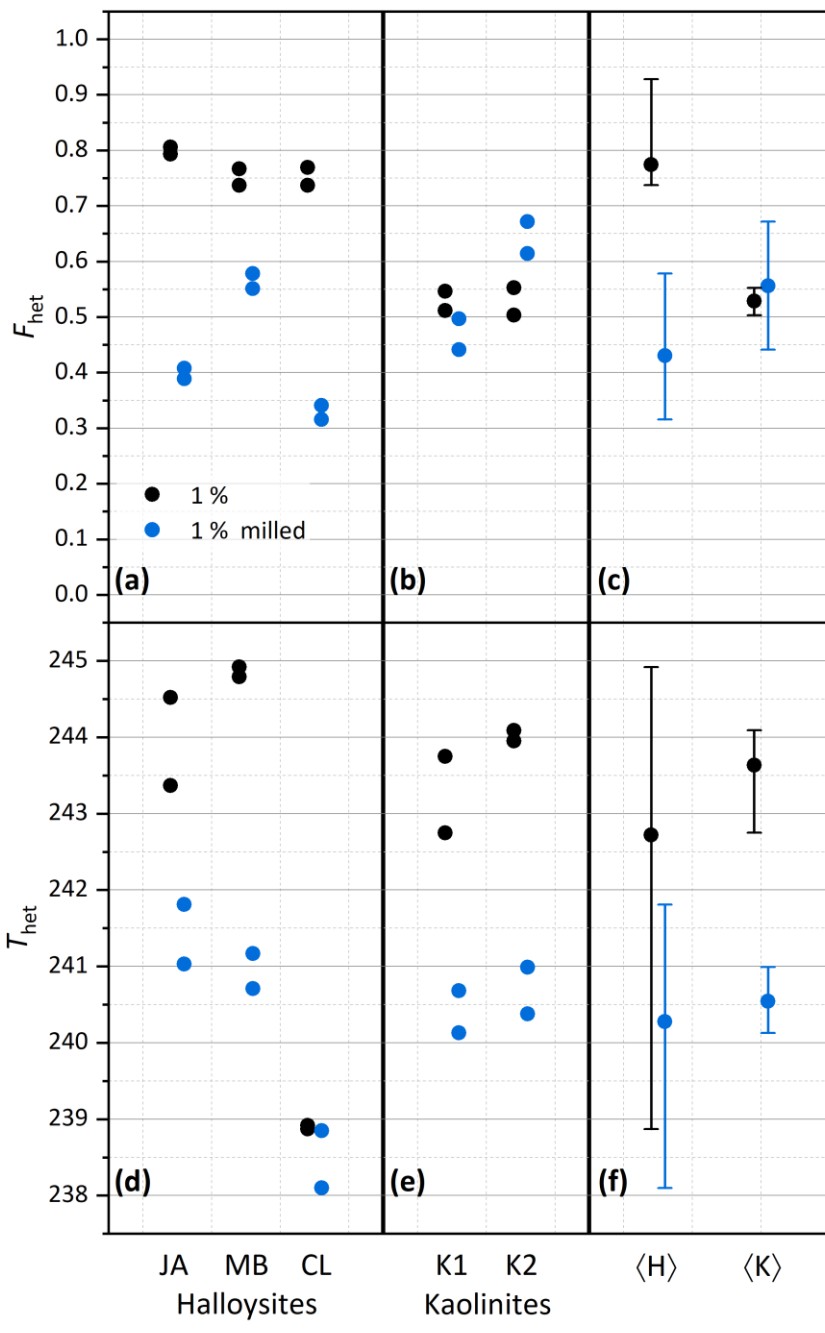

Figure 4: Summary of the effect of milling on heterogeneously frozen factions $F_{het}$ (a-c) and heterogeneous freezing onset temperatures $T_{het}$ (d-f) of 1 wt % halloysite and kaolinite samples (with sample codes identical to those in Fig. 1). Experiments have been performed for samples immersed without prior milling (black symbols) and immersed after prior milling (blue symbols). Each sample and milling procedure was measured twice. Parts (a) and (d): halloysites. Parts (b) and (e): kaolinites. Part (c) and (f): averages for the halloysites, ⟨H⟩, and for the kaolinites, ⟨K⟩, with vertical bars representing the spread (min–max) of all measurements.





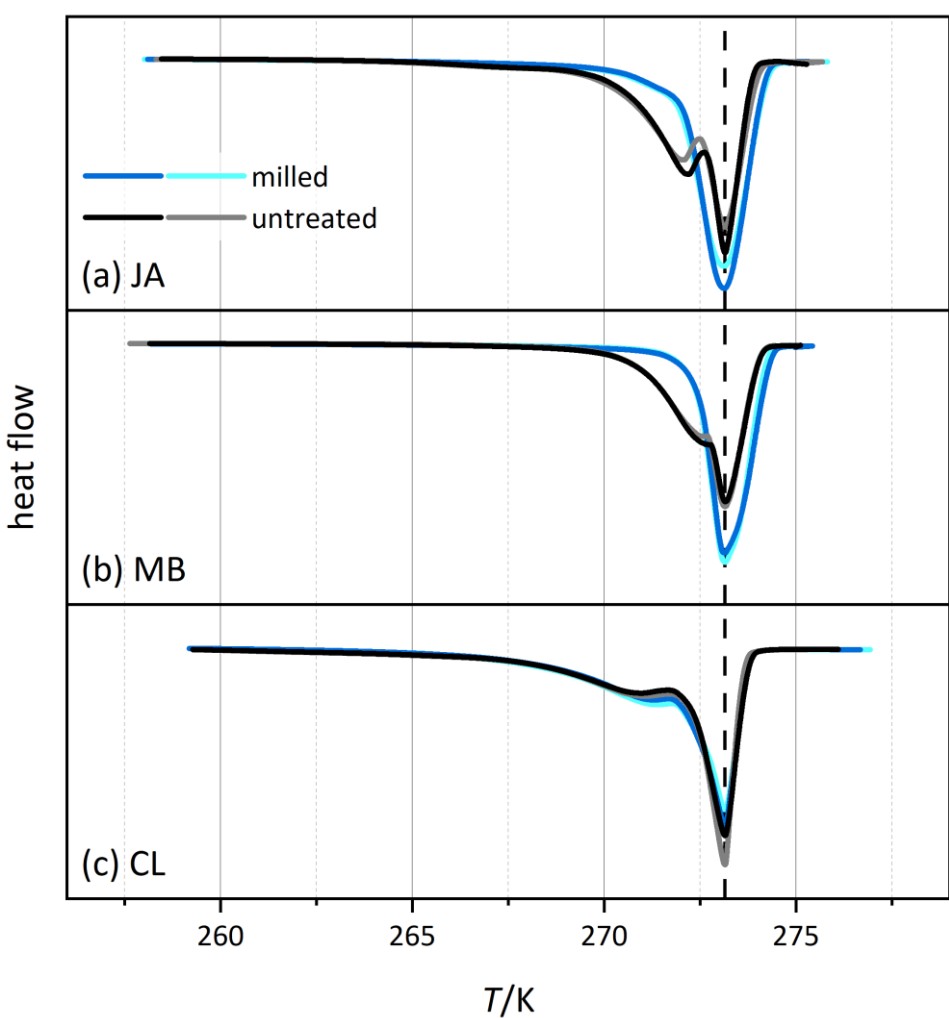


**Figure 5:** Thermograms from melting experiments of slurries of selected halloysite samples (black/grey) and the corresponding measurements of the milled samples (blue/cyan). The curves are normalized with respect to their total integral. Dust sample codes are identical to those in Fig. 1.





The pore volume distribution derived from the DVS measurements after milling shows a clear decrease for pore diameters
       > 10 nm, which we assign to the halloysite tubes, while the pore volume for diameters < 10 nm increases (Fig. 7). Only the
       CL sample still features a maximum in pore volume, which shifted slightly to larger diameters (17–20 nm) compared with the
       untreated sample (~10 nm). The samples JA and MB do not exhibit a maximum in the pore volume distribution for diameters
       > 10 nm after milling, which is in agreement with the melting peaks, which lack the second maximum due to pore ice melting
after milling. Interestingly, with this loss of pore volume, the IN activity of halloysites becomes similar to the one of the
       kaolinite samples.

       This shows that the tubular morphology strongly affects the IN activity. We therefore analyse in the following the relationship
       between morphology and IN activity in more detail.

       The TEM micrographs shown in Figs. 8 and 9 illustrate the morphologies of the untreated samples MB, JA, CL, and KGa-2
(K2), and the milled samples MB, JA, and CL, respectively. CL appears as the most homogeneous sample consisting almost
       exclusively of long narrow tubes with typical lengths and widths of 500–1000 nm and ~50 nm, respectively. The average inner
       diameter as determined by DVS is ~10 nm, thus, the average wall thickness should be ~20 nm. This homogeneity is reflected
       in the IN activity as CL exhibits a quite narrow heterogeneous freezing peak with onset slightly below 240 K. Quite narrow
       tubes are also dominating the sample JA, yet there is more diversity in length (10–1000 nm) and width (20–80 nm) including
some broader tubes like the one shown in panel (f). This larger diversity is reflected by the wider pore volume distribution in
       Fig. 6 with a peak at 20 nm that extends to 45 nm, and is in correspondence with a broader heterogeneous freezing signal in
       the DSC thermogram with a tail that extends up to ~243 K. The sample MB is even more heterogeneous than JA with "long
       and thin, short and stubby, tubular, spheroidal and plate-like" morphologies (Pasbakhsh et al., 2013). Accordingly, the pore
       volume distribution is very wide with two slight maxima at 17 and 45 nm in Fig. 6. Interestingly, this sample features a broad
heterogeneous freezing signal with onset at about 245 K and two maxima. Thus, there seems to be a relationship between the
       tube width and the IN activity. Yet, pores are not a prerequisite for IN activity as also the platy particles of kaolinite KGa-2
       (Fig. 8, panels (n–q) give rise to a heterogeneous freezing signal in the DSC with a quite high onset temperature, yet lower $F_{het}$
       than for the halloysites. Therefore, the tubes seem to have an indirect effect on IN activity by modulating the characteristic
       freezing temperature and the abundance of nucleation sites. An indirect influence of pores on IN activity is supported by Fig.
10, which shows that the pore volume neither correlates with $T_{het}$ nor with $F_{het}$, yet the decrease in pore volume due to milling
       (by up to 0.15 cm$^3$g$^{-1}$) goes along with a decrease in both, $T_{het}$ and $F_{het}$.

       To investigate more broadly the effect of milling, Fig. 11 correlates the IN activity with surface area before and after milling.
       An increase in surface area goes indeed along with an increase in $F_{het}$ for the untreated samples, with CL as the only exception.
       Yet, this correlation is destroyed when milled samples are included, as these exhibit lower $F_{het}$ despite increased surface areas.
Here again, CL is an exception, as it exhibits the highest surface area before milling, which slightly decreases after milling.
       The TEM micrographs shown in Fig. 9 further illustrate the effect of milling. In accordance with the DSC and DVS
       measurements, which retained the pore-water peak, the tubes of CL still appear intact after milling.



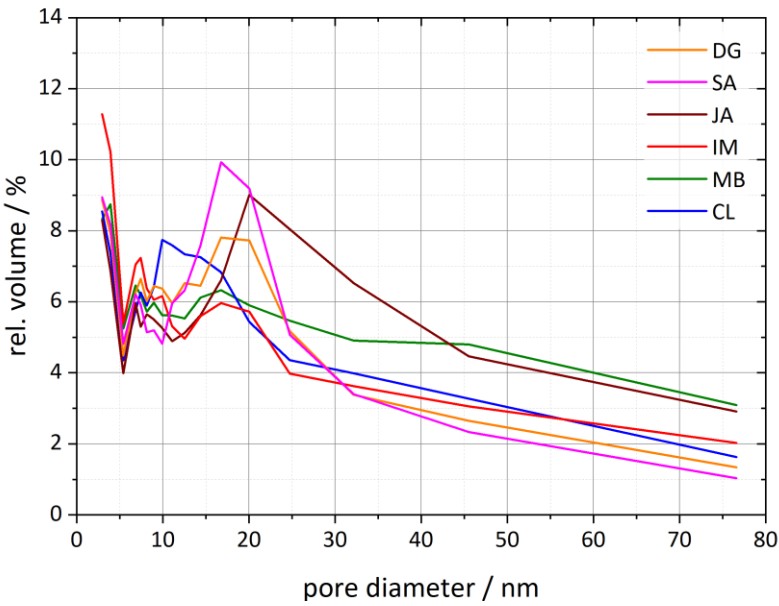

**Figure 6:** Pore size distributions for each of the six halloysite samples resulting from DVS isotherms analysed between 35 and 98 % RH.
Plotted is the relative pore volume against the pore diameter (calculated using Eq. 1 in Sect. 3.3). Dust sample codes are identical to those in Fig. 1.

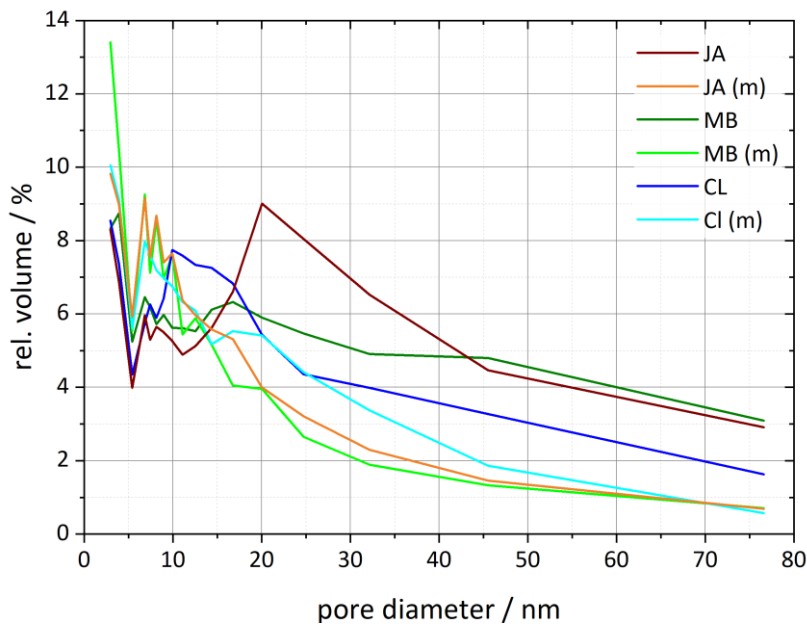

**Figure 7:** Pore size distributions for selected halloysite samples and their corresponding milled samples resulting from DVS isotherms analysed between 35 and 98 % RH (calculated using Eq. 1 in Sect. 3.3). Dust sample codes are identical to those in Fig. 1.






The untreated CL tubes show a tendency to form bundles, which seem to break up and disperse due to milling. For the JA and MB samples, the effect of milling is clearly visible: besides intact tubes, there are crushed tubes and debris present. These findings are in accordance with the DSC and DVS analyses, which witness a clear loss of pore water. After milling, there are more short tubes, thus there is also breaking of tubes.

The loss of IN activity through milling is in clear contrast to the findings for milled quartz, which shows an increase in IN activity after milling. This different impact of milling can be explained by the different structure of quartz compared with clay minerals. As quartz is built of a three-dimensional crystal lattice, breaking up of Si–O–Si bonds by vigorous milling is required to crush it. This generates radicals and reactive sites, which seem beneficial for ice nucleation (Kumar et al., 2019a). Instead, the layered structure of kaolinites and halloysites is easily crushed. Milling can lead to the breaking of layers or to the

displacement of layers with respect to each other.

**4.3 Location of ice nucleation**

Kaolinite and halloysite particles exhibit three different types of surfaces: the basal silica surface terminated by siloxane groups, the basal alumina surface and the edges, which both are hydroxylated (Schoonheydt and Johnston, 2006). In the following, we combine the findings of this study with the physicochemical properties of the surfaces to elucidate where ice

nucleation may take place.

In the absence of isomorphic substitution of $Si^{4+}$ by $Al^{3+}$, the tetrahedral sheets of kaolinite do not bear any charge and there are no surface-adsorbed charge-balancing ions required. As the siloxane surface of idealized kaolinite is uncharged and not hydroxylated, it is hydrophobic (Schoonheydt and Johnston, 2006). Using molecular dynamics simulations, Šolc et al. (2011) determined a contact angle of 105° for it. In real kaolinites and halloysites, there is some isomorphic substitution, which may

decrease the crystallinity of kaolinites and has also been hypothesized to cause kaolinite plates to roll up into halloysite tubes (Tarì et al., 1999). As the siloxane surface is not hydroxylated, it lacks a prerequisite for ice nucleation (Pedevilla et al, 2017). Nevertheless, Zielke et al. (2016) were able to grow ice on it in their molecular dynamics simulations. Yet, the siloxane surface as the location of ice nucleation cannot be reconciled with the high sensitivity of $T_{het}$ and $F_{het}$ of halloysites on the morphology. Different degrees in isomorphic substitution might account for differences in IN activity between halloysite samples, yet it is

not able to explain the changes due to milling since the degree of isomorphic substitution should not change by milling.

Several molecular dynamics studies attest the alumina surface the capability to nucleate ice (Zielke et al., 2016; Sosso et al., 2016; Glatz and Sarupria, 2018). Soni and Patey (2021) found that subtle differences in surface morphology may indeed explain the very low IN activity of the gibbsite alumina surface compared with the much higher one of kaolinites (Kumar et al., 2019b). As the alumina surface is the inner surface of the halloysite tubes, it may show relevant curvature that could

influence surface properties. Yet, there is no obvious correlation of IN activity with the curvature of the basal surfaces: the flat kaolinite platelets and the narrow tubes of the CL sample evidence both lower $T_{het}$ and $F_{het}$ than the MB sample, which consists





of a variety of different morphologies. Moreover, the layers of the halloysite tubes likely remain curved even after the tubes have been crushed as the curvature is induced by structural properties of the layers like isomorphic substitution.

**Figure 8:** Transmission electron microscopy (TEM) micrographs of untreated samples at two different magnifications (see length bars within the images. Dust sample codes are identical to those in Fig. 1.



**Figure 9:** TEM micrographs of milled samples at two different magnifications (see length bars within the images. Dust sample codes are identical to those in Fig. 1.




This leaves the edges as the most probable location for ice nucleation as has been suggested by Kumar et al. (2019b). The highly hydroxylated edges consist of OH–Si–O–Al–(OH)$_2^-$ and OH–Si–O–Al–OH$_2$ chains at near neutral conditions (pH 6.5), which may protonate or deprotonate depending on pH (White and Zelazny, 1988). Overall, edge sites have been found to contribute 30 % to the total surface area of KGa-1b (K1) and 18 % to the one of KGa-2 (Bickmore et al., 2002). In kaolinite

platelets, the edges have large lateral extensions, while their thickness depends on the number of layers that are stacked on top of each other. Particle edges are typically bevelled and not at right angles to the flake surface (Chakraborty, 2014). According to CNT, a nucleation site needs to provide a compact (e.g. circular) area that is large enough to host a critical ice embryo. Parameterizations of CNT indicate that this area is in the range of 10–50 nm$^2$, with larger areas required to nucleate ice at higher temperatures (Kaufmann et al., 2017). If we assume a high degree of hydroxylation as a criterion for ice nucleation,

several layers should be properly stacked to provide a large, hydroxylated surface area. As single layers are 7.1–7.2 Å thick, about five properly stacked layers are required to generate a hydroxylated edge area of about 10 nm$^2$. Inspection of the kaolinite (KGa-2) micrographs (Fig. 8) shows layers of different sizes irregularly stacked to platelets of varying thickness. This great variability is likely leading to sites of different sizes and geometries, which may explain the quite broad heterogeneous freezing peak of this kaolinite and kaolinites in general. In contrast, in halloysites, layers are rolled to tubes, which leads to straighter

edges providing coherently hydroxylated surfaces, which may explain the higher $F_{het}$ of halloysites, but also the larger diversity in $T_{het}$ and $F_{het}$ between samples depending on their characteristic tube width and wall thickness with larger edge areas allowing freezing at higher temperatures. Consistent with this interpretation, milling with a disc mill will crush the tubes and introduce dislocations between the layers. This reduces the area of coherently hydroxylated edge surfaces and would explain the similar $T_{het}$ and $F_{het}$ of milled halloysites and kaolinites.

The relevance of the number of layers stacked together for the IN activity of kaolin minerals is also supported by the finding that $T_{het}$ of KGa1-b and KGa-2 increased by about 4 K between the measurements by Pinti et al. (2012) and the present study, as storage at dry conditions can lead to the collapse of layers resulting in thicker stacks. The decrease in $T_{het}$ of these samples after one day in water can be explained by delamination that occurs during aging for extended times in water (Stçepkowska, 1990). Milling also changes the particles morphology. It may lead to folding and gliding of layers, particle size diminution

through breaking and crushing of layers, and aggregation (Stepkowska et al., 2001). Prolonged grinding decreases the crystallinity and ultimately leads to amorphization (Kristóf et al., 1993). Stepkowska et al. (2001) found a decrease in average particle thickness of KGa-1 already after milling for 1 min with an oscillatory mill: the average particle thickness at RH = 50 % decreased from 42.1 nm to 27.2 nm, followed by a further decrease to 13.6 nm after 5 min due to delamination. Additional milling led again to a slight increase in particle thickness to 15.6 nm probably due to aggregation. For our very short milling

time of 10 s, delamination should prevail. A decrease in average particle thickness can explain the observed decrease in $T_{het}$ due to milling, if the nucleation temperature is indeed limited by the spatial extension of the nucleation sites.





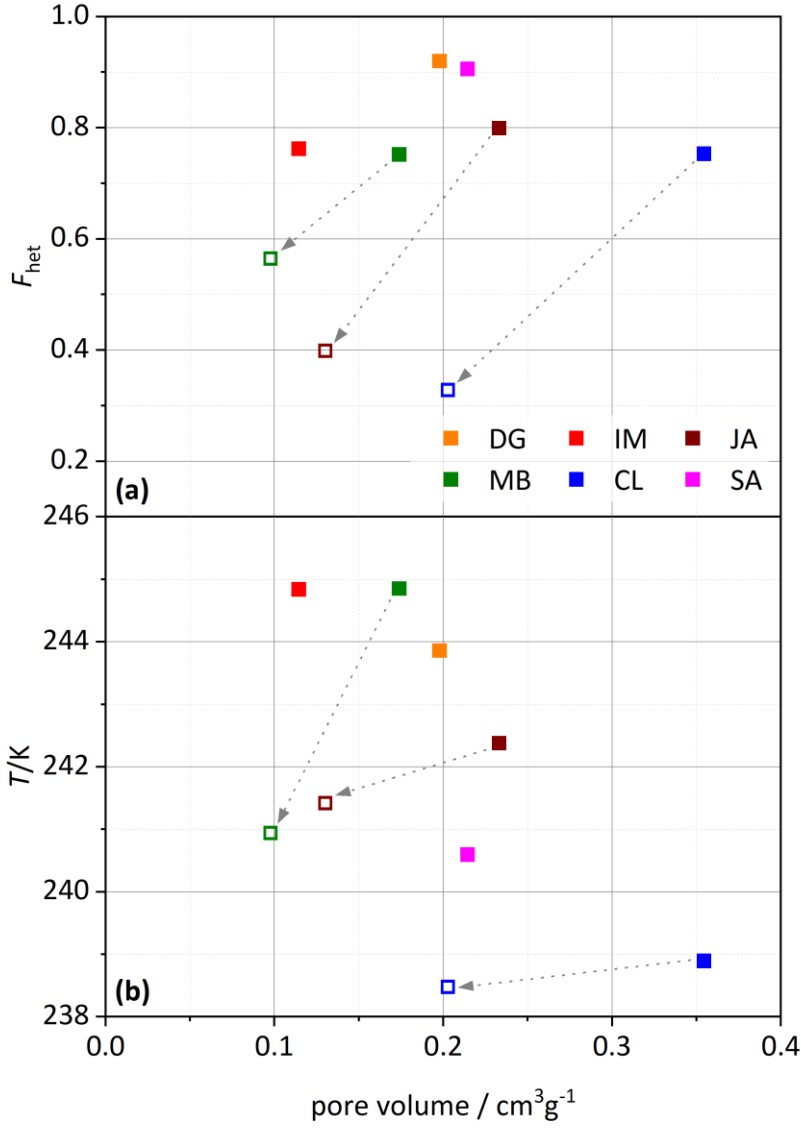

**Figure 10:** (a) Heterogeneously frozen fraction, $F_{het}$, and (b) heterogeneous freezing onset temperature, $T_{het}$, plotted against the specific pore volume calculated from DVS measurements (analysed between 40–98 % RH). Filled symbols depict untreated and open symbols milled samples. The dashed arrows point from the untreated to the corresponding milled sample. Dust sample codes are identical to those in Fig. 1.





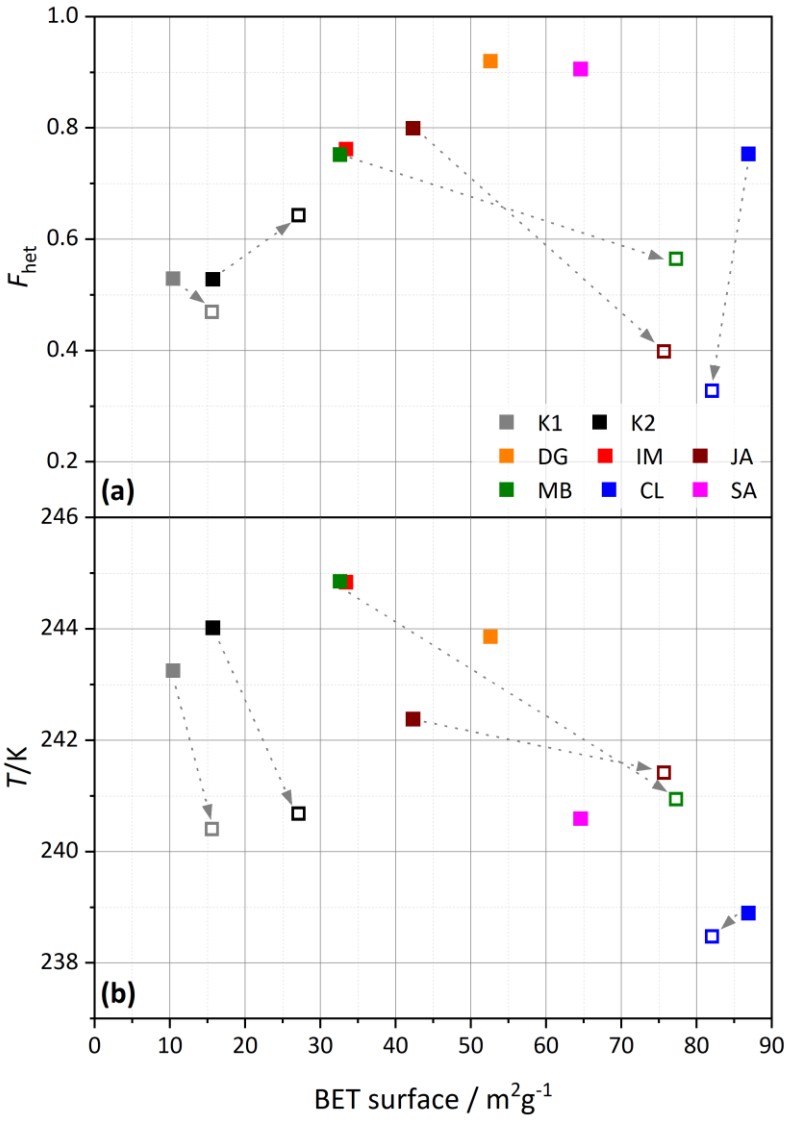

**Figure 11:** Similar as Fig. 10, (a) Fhet and (b) Thet, but here plotted against the BET surface area derived from DVS measurements (analysed between 15 – 35 % RH). Filled symbols depict untreated and open symbols milled samples. The samples are colour coded as indicated by the legend above the top panel. The dashed arrows point from the untreated sample to the corresponding milled sample. Dust sample codes are identical to those in Fig. 1.



Conversely, the increase in $F_{het}$ of KGa-2 after milling may be explained by an increase in edge surface area through breaking and crushing of particles.

Overall, the finding that montmorillonites, kaolinites, and feldspars all exhibit IN activity that is enhanced in dilute ammonia or ammonium solutions points to chemical similarities in their nucleation sites. The surface structure that they have in common is the OH–Al–O–Si–OH functionalized surface, which is the only surface type in feldspars. Thus, only ice nucleation at the edges of clay minerals can explain the increase in IN activity as a common feature of aluminosilicates. Kumar et al. (2022), who addressed this question by investigating smectites, also identified the edges as the key feature to explain observed

variations in IN activity.

## 5. Conclusions

Immersion freezing experiments of emulsified dust suspensions of two kaolinite and six halloysite samples illustrate the different ice nucleation properties of these chemically identical but morphologically different kaolin minerals. The predominantly tubular halloysite samples yield a higher diversity in freezing behaviour compared to the two kaolinite samples,

visible in the heterogeneous freezing onset temperatures, the heterogeneously frozen fraction, and the overall shape of the thermograms. To elucidate the role of particle morphology, selected samples were milled. Freezing experiments performed after milling yielded significantly reduced IN activity for the halloysite samples, but hardly any change for the kaolinite samples. A reduction in IN activity upon milling contrasts with the enhancing effect that milling has on the IN activity of quartz. The freezing experiments were complemented with DVS measurements, pore ice melting experiments performed with

slurries, and TEM micrographs before and after milling. For almost all samples the 10 s milling process is accompanied with an increase in specific surface area. Additionally, all halloysite samples experience a loss in specific pore volume, which indicates the demolition of tubes due to milling. The concomitant decrease in $F_{het}$ and in $T_{het}$ shows that the tubular morphology of the halloysite samples influences their IN activity.

Based on the comparison of the IN activity of kaolinites and halloysites with the one of aluminosilicates in general, and in

combination with surface chemical and geometrical arguments, we conclude that among the three surface types present in kaolin minerals, the hydroxylated particle edges are the most probable location for ice nucleation:

- As particle edges are hydroxylated, they have the capacity to form hydrogen bonds with water molecules, which is often considered a prerequisite for ice nucleation.

- The OH–Al–O–Si–OH functionalized edges are the surface structure that kaolin surfaces have in common with

montmorillonites and feldspars, which all exhibit IN activity that is enhanced in the presence of ammonia and ammonium containing solutions.

- Particle edges of kaolin minerals exhibit a high diversity in geometry that can account for the highly diverse IN activity observed for the different halloysites, whereas the basal surfaces are topologically rather uniform. We assume that the



freezing temperature is the higher the larger the surface area spanned by the hydroxylated edges is. The dimension of this area depends on the number of stacked layers and on their stacking order. Particle edges are more likely to be properly stacked on top of each other in the tubular morphology of the halloysites since the freedom of lateral displacement between rolled layers is limited. This would explain the on average higher IN activity of the halloysites compared with kaolinites as well as the differences among the halloysite samples because of their different characteristic tube dimensions.

*Data availability.* The data presented here will be made available on www.research-collection.ethz.ch

*Author contributions.* KK and AAH conducted the experiments. KK, AAH, CM, and TP contributed to the planning and interpretation of the experiments. KK prepared the manuscript with contributions from CM and TP.

*Competing interests.* The authors declare that they have no conflict of interest.

*Acknowledgements.* We acknowledge the Swiss National Foundation for financial support (project numbers: 200021_175716). We thank Michael Plötze, Anette Röthlisberger, and Marion Rothaupt for XRD measurements; Christoper Dreimol providing us with DVS measurements; Marco Griepentrog for milling; Eszter Barthazy of ScopeM for support and assistance related to TEM imaging; Ulrich Krieger, Uwe Weers, and Marco Vecellio for support in the laboratory (all ETH). We thank Silvan von Arx from the Institute of Mechanical Engineering and Energy Technology (Lucerne School of Engineering and Architecture, Lucerne) for providing size distribution measurements (Laser diffraction particle sizer). We thank Pooria Pasbakhsh for providing the halloysite samples and valuable insights into the world of halloysite.



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
