# Peer review of "Comparing the ice nucleation properties of the kaolin minerals kaolinite and halloysite"

_Atmospheric Chemistry and Physics, 2022_

## Author Comment (AC1)

**We thank Reviewer 1 for his/her thoughtful comments. We reproduce the reviewer's comments in black and our responses in blue. Line numbers refer to the revised, marked up manuscript.**

In the presented study the ice nucleation properties of two types of kaolin minerals are investigated, which are chemically identical but have different morphologies. While kaolinite forms flat platelets and has a more constrained ice nucleation behaviour (e.g., freezing onset temperature in the range of 243.3 K to 244 K using freshly prepared samples), halloysite has a variety of different morphologies, e.g. tubes, and shows a more diverse ice nucleation activity with ice onset temperatures ranging from 238.2 K to 244.9 K. To better understand the role of morphology, the samples are milled and reveal a clear decrease in the ice nucleation ability of halloysite, while kaolinite samples are rather unaffected. By determining the pore size distributions and pore volumes of the samples before and after milling, it is shown that the halloysite tubes are destroyed, and thus it is suggested that they are likely involved in ice nucleation processes. The authors provide a detailed discussion about the surface type of the mineral causing the ice formation and conclude that hydroxylated particle edges are the most likely location for ice nucleation.

The study is well conceived and I enjoyed reading the manuscript which is generally very well written. I only have minor comments.

**General comments:**

Do you have suggestions for further studies to test your hypothesis that the hydroxylated edges of the kaolin minerals are causing the ice nucleation, e.g., molecular dynamics studies, or other laboratory studies?

> Molecular dynamics would certainly be an interesting option. Usually, molecular dynamics studies rely on the regular surface lattice of the respective mineral as the starting point for simulations of ice nucleation. The structure of the edges, in contrast, are not well defined, and the specific features that serve as nucleation sites are unknown. Therefore, in a first step, different surface edge structures would need to be defined before their ability to nucleate ice could be assessed.

> As direct experimental observation of the nucleation process is not feasible, only circumstantial evidence can be gathered. To this end, the IN activity of still other clay minerals could be assessed and correlated with surface and stacking properties, as we have already done in the recent study by Kumar et al. (2022), where we correlate the IN activity of different montmorillonites with exchangeable cations and stacking thickness. This study is still in discussion in Atmos. Chem. Phys. Discuss.

Abstract: You might want to consider mentioning that the milling leads to an increase in specific surface area.

> We modify the abstract starting from line 17:

> "To interpret these findings, the freezing experiments were complemented by dynamic vapour sorption, BET (Brunauer, Emmett, Teller) surface area measurements, pore ice melting experiments with slurries, and transmission electron microscopy (TEM) before and

after milling. These measurements demonstrate an increase in surface area and the destruction of tubes by milling …. "

Line 65: An early study by Vonnegut (1947) should be referenced here as well.

Added

Section 2: It might be helpful to include a figure showing the structure of kaolinite and halloysite.

We added a figure that illustrates the kaolinite and halloysite structures as the new Fig. 1 and renumbered the other figures accordingly.

Lines 182 and 197: Shouldn't other studies next to Klumpp et al. (2022) be referenced here as well?

We added additional references on line 177 of the revised manuscript as suggested by the reviewer.

Lines 210 to 211: Is there a reason why those halloysite samples were chosen for milling (e.g., the content of impurities)?

We chose the halloysite samples to cover different morphologies and different DSC curve types of the untreated samples. The amount of halloysite sample available also played a role.

Lines 230 to 232: Can you explain in more detail this equation, and also provide uncertainty estimates for your measurements for pore volume distributions?

For a detailed explanation of the equation, we refer to Kocherbitov and Alfredsson (2007) and their references in the manuscript. Unfortunately, uncertainty quantification is impossible given the different assumptions incorporated in Eq. 1. However, we are confident that for the purpose of comparison between the samples measured and analysed in this study the relative uncertainty is small since the systematic uncertainties are kept constant.

Line 278: The description of the experiments using ammonia/ammonium is missing in the methods.

We added a reference to the experiments performed in ammonia solution by revising the sentence starting on line 185:

"Suspensions of kaolinite and halloysite with 0.2 and 1 wt % in pure water (molecular bioreagent water, Sigma Aldrich) or in 0.2 M ammonia solution (prepared from Merck 25 % ammonia aqueous solution) were prepared and sonicated for 5–10 minutes. "

Line 358 and following: You might want to consider referencing Fig. 2 here.

Following the reviewer, we added the reference to Fig. 2 on line 383 after "240 K" by adding "see Fig. 2", on line 387 after "~243 K", and on line 391 after "maxima".

Section 4.3: I recommend naming this section slightly differently, to indicate that this is a discussion and not a results section (e.g., "likely location of ice nucleation").

> We follow the suggestion of the reviewer and rename the section "Likely location of ice nucleation".

Figs 7, 8, 11 and 12: Could you indicate the uncertainties in your measurements by error bars?

> Following the reviewer, we add error bars for temperatures and $F_{het}$. Given the high precision of the DVS technique (explained in Section 3) and the difficult treatment of systematic uncertainties we do not indicate error bars for values derived from DVS.

**Technical comments:**

Line 3: "1" is missing in the authors' name for their affiliation.

> Added.

Figs. 4 and 5: While in Fig. 4 the untreated samples are labeled "pure", there are not specifically labeled in Fig. 5.

> We changed to "untreated" in both cases.

Lines 446 and 448: A bracket is missing at the end of the sentence.

> Bracket added.

**Reference**

Vonnegut, B.: The Nucleation of Ice Formation by Silver Iodide, J. Appl. Phys., 18, 593-595, 10.1063/1.1697813, 1947.

---

## Author Comment (AC2)

**We thank Reviewer 2 for his/her thoughtful comments. We reproduce the reviewer's comments in black and our responses in blue. Line numbers refer to the revised, marked-up manuscript.**

**General comments:** The extensive work by Klumpp et al. describes the heterogeneous ice nucleation properties of two mineral dust types, kaolinite and halloysite, that are chemically identical and have the same crystal structure, but differ in their morphologies. Comparison of their freezing properties before and after physical modification by milling showed that the morphology plays an important role affecting the chemistry as well. This study has an important contribution to the ice nucleation field of study and it improves our understanding of why a certain surface produce better ice nucleating sites compare to another. The manuscript is within the scope of ACP. The experiments were well designed, the results are well presented and well interpreted. The introduction is very thorough and focused. I recommend to publish the manuscript in ACP after the authors will address the following comments:

**Major comments:**

1. The authors suggest the notion of surface hydroxylated edges as ice nucleating sites, however, the current work did not provide a direct evidence for that. So, there is no mechanism which was found and I think this should be highlighted in the conclusion and discussed. Is there scientific way to establish this for example? A way to quantify that?

   This is indeed a good question. Unfortunately, as long as direct experimental observation of the ice nucleation process is not feasible, we are left with circumstantial evidence to derive the role of the edges in ice nucleation on clay minerals. One major obstacle to establish a mechanism for ice nucleation is the unresolved chemical and topographic makeup of nucleation sites. These are typically defects in the regular mineral surface, which are too small and too few to be characterized experimentally.

   To address this topic, we add the following text to the revised manuscript at the end of the discussion section on line 501:

   "Yet, such comparative, experimental studies cannot elucidate the microscopic mechanism of heterogeneous ice nucleation. Specifically, they are unable to explain how topographical features and surface functional groups play together to boost the probability of ice nucleation above the one of bulk water at the same temperature. Traditionally, a direct templating effect is assumed to explain the IN activity of a surface. However, such a direct orientation of water molecules into an ice-like pattern seems unlikely given the irregular structure of the clay mineral edges. Rather, the requirement of a sufficient spatial extension of the nucleation site could point to a rearrangement of the liquid water structure to a higher degree of hydrogen bonding within a water volume of the size required to host the critical ice embryo. The role of the surface would then be to induce such a rearrangement by disrupting the prevalent liquid water structure at that temperature through hydrogen bonding to water molecules."

2. What are the atmospheric implications? kaolin particles are common in the atmosphere and will atmospheric transport or cloud processing can affect its ice nucleating abilities in light of the results of this study?

This is again a very interesting question, which we address in the revised manuscript by inserting the following paragraph at the end of the discussion section starting from line 510:

"This study shows that freezing temperatures of kaolin particles are limited by platelet thickness, i.e. the number of layers stacked together in a particle. Thus, larger particles should be able to freeze water at higher temperatures than smaller ones. Indeed, bulk measurements with larger volumes show freezing temperatures up to or even above 260 K, potentially occurring on supermicron particles (Zimmermann et al., 2008; Pinti et al., 2012; Whale et al., 2015). Moreover, if immersion in water led to a partial delamination of the layers, a decrease in freezing temperature depending on the freezing mode could ensue. Indeed, there is experimental evidence that freezing temperatures are higher in contact and condensation mode, where dry particles induce freezing, than in immersion mode, where ice nucleation starts from wetted particles (Zimmermann et al., 2008; Lüönd et al., 2010; Welti et al., 2014; Whale et al., 2015; Nagare et al., 2016)."

3. In many cases along the manuscript the data presented without the uncertainties. Please make sure you report the uncertainties.

Following the reviewer(s) we add error bars/uncertainties when possible (Figs. 11 and 12). Please note the newly added section about the DVS method where we also mention the uncertainties (stop criterion: mass change rate ≤ 0.0005 % per minute) connected to the method. Given this precision we do not display error bars on data derived from DVS measurements.

4. There is no information about sample preparation and measurement process for surface area in the BET. Degassing was done? At which conditions? How many times each powder was measured?

Following the reviewer, we extended Section 3.3. Now it contains a detailed description of the DVS measurements.

**Minor comments**:

1. A figure describes kaolinite and halloysite structures can be included in the text or in the supplementary.

We added a figure that illustrates the kaolinite and halloysite structures as the new Fig. 1 and renumbered the other figures accordingly.

2. What is the source of difference between results of $H_2O$ and $N_2$ surface area and pore size?

Surface adsorption of $H_2O$ depends on the hydrophilicity of the surface. In the case of completely hydrophobic surfaces, there would be no adsorption of water vapour at all. The rather hydrophobic siloxane surface of kaolin minerals could lead to a low bias in surface area. Conversely, pores with diameters below 2 nm would fill with water below a water vapour saturation ratio of 0.35 and therefore interfere with the determination of surface area. Such pore space could arise at the rugged edges of kaolinite and halloysite, thus inducing a high bias in the surface area determined through water vapour adsorption. We consider therefore the agreement between $H_2O$ and $N_2$ surface areas as astonishingly good. It should also be noted that DVS measurements take place at 25°C compared with -196°C for $N_2$-BET analysis, which could affect the total surface area, too.

3. Line #271: Please detail to which sample variability you refer to?

We reformulate to:

"The smaller differences between repetitions for the higher concentrated samples suggests that the larger portion of sample investigated at 1 wt % represents the average sample properties better than the five times smaller portion required at 0.2 wt %, which would imply composition or particle size inhomogeneity within the sample.

4. Line #275: Please explain what is the cause for the 1 K difference in the homogeneous freezing temperature between the different measurements.

Since the heterogeneous and homogeneous peaks overlap to different extent for every sample, the homogeneous onset cannot be determined to high accuracy. We mention the homogeneous freezing temperature in the text to provide the reader with a point of orientation within the DSC curves rather than to draw any conclusion from its value or spread.

**Technical comments**:

Line #86: Please provide a reference.

We added Murray et al. (2012), Boose et al. (2016) and Kaufmann et al. (2016)

Line #192: For uniformity, notice the use of backslash in temperature change rate here while in the rest of the text there is a use of superscript.

Changed to superscript

Fig 3: missing units to the y-axis for Thet.

Added

Fig 5: missing units to the y-axis for Thet.

Added

Fig 9. Missing bracket in the caption.

Added

Fig 10. Missing bracket in the caption.

Added

**References**

Lüönd, F., Stetzer, O., Welti, A., and Lohmann, U.: Experimental study on the ice nucleation ability of size-selected kaolinite particles in the immersion mode, J. Geophys. Res. Atmos., 115, https://doi.org/10.1029/2009JD012959, 2010.

Marcolli, C., Gedamke, S., Peter, T., and Zobrist, B.: Efficiency of immersion mode ice nucleation on surrogates of mineral dust, Atmos. Chem. Phys., 7, 5081–5091, https://doi.org/10.5194/acp-7-5081-2007, 2007.

Nagare, B., Marcolli, C., Welti, A., Stetzer, O., and Lohmann, U.: Comparing contact and immersion freezing from continuous flow diffusion chambers, Atmos. Chem. Phys., 16, 8899–8914, https://doi.org/10.5194/acp-16-8899-2016, 2016.

Welti, A., Kanji, Z. A., Stetzer, O., Lohmann U., and Lüönd, F.: Exploring the mechanisms of ice nucleation: From deposition nucleation to condensation freezing, J. Atmos. Sci., 71, 16–36, https://doi.org/10.1175/JAS-D-12-0252.1, 2014.

Whale, T. F., Murray, B. J., O'Sullivan, D., Wilson, T. W., Umo, N. S., Baustian, K. J., Atkinson, J. D., Workneh, D. A., and Morris, G. J.: A technique for quantifying heterogeneous ice nucleation in microlitre supercooled water droplets, Atmos. Meas. Tech., 8, 2437–2447, https://doi.org/10.5194/amt-8-2437-2015, 2015.

Zimmermann, F., Weinbruch, S., Schütz, L., Hofmann, H., Ebert, M., Kandler, K., and Worringen, A.: Ice nucleation properties of the most abundant mineral dust phases, J. Geophys. Res., 113, D23204, https://doi.org/10.1029/2008JD010655, 2008.